# Confucian Cosmopolitanism: The Modern Predicament and the Way Forward

**Ruihan Wu**

Department of Philosophy and Science, Southeast University, Nanjing 210009, China; wuruihan@seu.edu.cn

**Abstract:** In the Chinese-speaking academic community, the topic of Confucian cosmopolitanism is intricately linked to the concepts of "Tianxia" and "Datong", carrying significant political implications. This context arises from the tension between the Confucian vision of a borderless world order and the reality of the bounded nation-state system since the late 19th century. This modern situation constitutes the dual predicaments for Confucian cosmopolitanism: the contradiction between the logic of Datong and the logic of national empowerment, as well as the conflict between the specific Confucian identity and the universal concern for the world. Represented by notable figures like Liang Qichao, modern scholars have devoted themselves to resolving these predicaments. On one hand, Liang, in contrast to his teacher Kang Youwei, emphasized the coexistence of the global ideal and the nation-state system. He proposed the concept of a 'cosmopolitan nation,' which not only considers nationalism as a stepping stone toward cosmopolitanism but also views the nation as an organizational form with the world as its ultimate purpose. This response addresses the first predicament. On the other hand, Liang redirected the focus of cosmopolitanism to the individual, establishing a connection with the core Confucian value of Ren. He interpreted the ideal of Datong as the awakening and refinement of each individual's kinship consciousness, thereby mitigating the constraints imposed by Confucian identity and the national narrative on the discourse of cosmopolitanism. This tackles the second predicament. Reflecting on these modern predicaments not only sheds light on the political reasons underlying Confucian cosmopolitanism but also reveals its broader dimension as a universal ethical concern.

**Keywords:** Tianxia; Datong; nationalism; modernity; Liang Qichao; kinship consciousness



## 1. Introduction

In the Western philosophical tradition, "cosmopolitanism" is a multifaceted and richly nuanced topic with roots in various historical sources. From Diogenes' concept of "kosmopolitēs" to the Stoic idea of a "world city-state" and Kant's vision of a "league of nations", each of these intellectual origins sheds light on different issues. In contemporary philosophical discussions, the exploration of cosmopolitanism has also diversified, giving rise to varied perspectives. For many philosophers, cosmopolitanism is regarded as an ethical matter, sparking debates about whether individuals should extend their care and assistance to all human beings without distinction or if they have specific duties primarily to their fellow compatriots. On the other hand, a significant number of thinkers view cosmopolitanism as a pressing political philosophical issue, grappling with questions about the most just and equitable form of global political organization (see Scheffler 1999; Kleingeld and Brown 2019).

Meanwhile, the issue of "Confucian cosmopolitanism" seems to be confined within the domain of ethics and moral philosophy. After the publication of Martha Nussbaum's essay "Patriotism and Cosmopolitanism" (Nussbaum 1994), Confucian scholars, whether explicitly or implicitly, have engaged with Nussbaum's ideas and focused on defining the characteristics of a Confucian cosmopolitan and how they can be cultivated. They argue that Confucianism, with its morally centered philosophy of Ren 仁 (humanity) and its

metaphysical emphasis on the Dao 道 (the way) and Tian 天 (heaven), provides a solid philosophical foundation for cosmopolitanism. These scholars perceive the ideal Confucian cosmopolitan as someone who embodies moral excellence and virtue, exemplified by the junzi 君子 (noble person) or sage (Neville 2012). Furthermore, they contend that Confucianism offers a distinctive form of "rooted cosmopolitanism" that addresses Nussbaum's concerns about global homogeneity (Peng 2023). They primarily view cosmopolitanism as an ethical or moral issue rather than a purely political one (see Ivanhoe 2014).

However, the situation is precisely the opposite in the Chinese-speaking academic community, particularly after Chinese leaders proposed the concept of a "community of shared future for mankind". The focus shifts towards studying a Chinese version of a world system that transcends the nation-state framework. Cosmopolitanism, as the translation of "shijie zhuyi" 世界主义, becomes closely associated with international political issues. Chinese philosophers, especially those studying Confucianism, feel the responsibility to seek the roots of a Chinese conception of the world in ancient philosophical resources. Therefore, concepts such as Datong 大同 (great harmony) from pre-Qin texts and the cosmological worldview of the unity of all things developed during the Song and Ming dynasties, along with the idealized view of Tianxia 天下 (all under heaven) and the tributary system in reality, are repeatedly mentioned and compared to the current global situation[1]. To some extent, Confucian cosmopolitanism now becomes the modern expression of the concept of "Tianxia".

The significant differences between the Chinese and English academic communities regarding Confucian cosmopolitanism prompt us to examine the context from a century ago. Cosmopolitanism was introduced to China during the late 19th century, and in the 1920s, Chinese intellectuals witnessed discussions on cosmopolitanism that went beyond the nation-state framework. In a radically different context from the pre-modern era, intellectuals influenced by tradition expanded and reconstructed the Confucian understanding of world order. This historical backdrop helps explain the semantic differences in the concept of cosmopolitanism. Nationalism as a rival engendered the modern predicament of Confucian cosmopolitanism, highlighting its political implications. However, modern Confucian intellectuals, like Liang Qichao 梁启超 (1873–1929), offered ways to transcend this predicament by reinterpreting the notions of Tianxia and Datong. Moreover, this approach reveals the commonalities between the two discourses of Confucian cosmopolitanism in China and the West.

## 2. Cosmopolitanism in Confucian Context: Tianxia, Datong, and the Boundary between Yi and Xia (夷夏之辨)

The term "Tianxia" has been widely recognized by Chinese scholars as the word used in ancient China to refer to the world. Its literal meaning suggests a close association with the concept of heaven in ancient Chinese thought. Although the notions of tian, tianming 天命 (heavenly mandate), and tiandao 天道 (the way of heaven) have exhibited quite multifaceted and evolving meanings within various philosophical schools, such as Confucianism, Daoism, and Mohism, the prevalent use of the term "Tianxia" unequivocally indicates that the ancient Chinese perceived the world through a lens of unity and comprehensiveness. As summarized by contemporary Chinese scholar Zhao Tingyang 赵汀阳, who holds significant influence in the discourse on the "Tianxia System", Tianxia is a philosophical construct encompassing geographical, psychological, and sociopolitical dimensions. Functioning as the bedrock of ancient Chinese comprehension of both the subjective and objective realms, "Tianxia" embodies a holistic worldview that integrates both humanistic and physical aspects, signifying a transition from chaos to "kosmos". It represents an "institutionalized world" that encapsulates the full concept of the world (Zhao 2011, p. 28).

Aligned with the systemic nature of "Tianxia" is the notion of "Datong". The treatises on Xiaokang 小康 (a state of moderate prosperity) and Datong found in the chapter "Li Yun 礼运" (Ritual Operation) of the Confucian classic the *Liji* 礼记 (*Book of Rites*) have formed

the cornerstone of the discussion on Confucian political philosophy in modern times. It says the following:

> When Great Dao was pursed, a public and common spirit ruled all under the Heaven. They chose men of talents, virtue, and ability; their words were sincere, and what they cultivated was harmony. Thus men did not love their parents only, nor treat as children only their own sons. A competent provision was secured for the aged till their death, employment for the able-bodied, and the means of growing up to the young. They showed kindness and compassion to widows, orphans, childless men, and those who were disabled by disease, so that they were all sufficiently maintained. Males had their proper work, and females had their homes. (They accumulated) articles (of value), disliking that they should be thrown away upon the ground, but not wishing to keep them for their own gratification. (They laboured) with their strength, disliking that it should not be exerted, but not exerting it (only) with a view to their own advantage. In this way (selfish) schemings were repressed and found no development. Robbers, filchers, and rebellious traitors did not show themselves, and hence the outer doors remained open, and were not shut. This was (the period of) what we call "Datong". (Zheng and Kong 1999, pp. 658–59)[2]

According to the views put forward by Kang Youwei 康有为 (1858–1927) and other modern intellectuals, the Confucian classics contain a series of political norms that are considered relevant to the era of Xiaokang. However, the authentic manifestations of Confucius' Datong doctrine are rarely encountered in the classical texts or historical records. The preceding passage depicts the multifaceted ethical relationships inherent in an ideal Datong society, accentuating its guiding principle, namely, "gong 公" (publicity). It advocates not only caring for one's own children but also ensuring that everyone receives care, not just accumulating wealth but utilizing resources to their fullest. These statements underscore the dissolution of boundaries between finite communities and the dichotomy of self and the others. While the transcendence of national borders is not explicitly addressed in this passage, subsequent descriptions of Xiaokang mention that "their object is to make the walls of their cities and suburbs strong and their ditches and moats secure" (Zheng and Kong 1999, p. 660), suggesting that Datong indeed implies the transcendence of political boundaries, encapsulating a world order at the scale of "Tianxia".

"Tianxia" is a comprehensive concept that encompasses everything, leaving no external world beyond its scope. This characteristic likely explains why Chinese thought does not generate concepts similar to "heresy", which is related to the trend of nationalism prevalent in the West. "Since there is no external world, Tianxia consists solely of an internal realm without incompatible externals, but rather internal structural relationships of proximity and distance. While China, like any other region, naturally develops localism centered around itself, it lacks clearly defined and universally applicable 'the Others,' as well as the consciousness of irreconcilable differences and the nationalism that demarcates boundaries from 'the Others.'" (Zhao 2011, p. 35). However, this viewpoint is a subject of ongoing debate. Prasenjit Duara has brought attention to the fact that the concept of Tianxia as a form of "culturalism" (as termed by Levenson) is not the only representation of community within the Confucian tradition. He highlights: "at least two representations of political community in imperial Chinese society are discernible: the exclusive Han-based one founded on an ascriptive principle and the another based on the cultural values and doctrines of a Chinese elite" (Duara 1996, pp. 59–60).

Confucianism indeed distinguishes between the internal and external aspects of the world. Throughout history, whenever nomadic tribes invaded the central empire, the concept of "Yixia 夷夏" from pre-Qin classics like *Chunqiu Gongyang Zhuan* 春秋公羊传 (*The Gongyang Commentary on the Spring and Autumn Annals*) would be invoked to reinforce the boundaries between the Han ethnic group and minority ethnic groups. *Gongyang Zhuan* states, "In *Chunqiu*, the state of Lu 鲁 is regarded as the 'internal,' while the states of other Xia ethnicities are considered the 'external.' Furthermore, within the Xia ethnicities, they

are seen as the 'internal,' whereas the barbarian tribes are seen as the 'external.' Since the king aims to unify Tianxia, why does the rhetoric of this classic text still differentiate between the 'internal' and 'external'? This is because unification should commence from the closest proximity" (Gongyang et al. 1999, pp. 400–1). The distinction between the "internal" and "external" in this context refers to the record of a gathering, where *Chunqiu* extensively documented the names of participants from feudal states in the central empire, but the remote state of Wu 吳 was only mentioned as "Wu", with the names of the attendees from that region omitted.

While this emphasis on the distinction between "Yi" and "Xia" may appear as racism, interpretations by successive Confucian scholars have shown that this distinction is compatible with the concept of Tianxia. *Gongyang Zhuan* already provides the progressive scheme of "commencing from the closest proximity". Furthermore, sanshi shuo 三世说 (the theory of three eras), attributed to Confucius, underwent progressive refinement by Confucian scholars to incorporate the issue of differentiating between the internal and external realms, shifting the concern of extending or eliminating the boundaries of "Yi" and "Xia" from a spatial problem to a temporal one. In times of extreme disorder, it was necessary to differentiate one's own state even from other states of the same ethnicity. As times improved, the focus shifted to distinguishing the Xia ethnic group from others. Ultimately, in the perfected world, all races should be treated equally without distinguishing proximity or distance. Tianxia, as a fully institutionalized world where the order of Datong can be applied, represents this third world. However, it is usually an ideal rather than a reality. Thus, Duara argues: "The universalistic claims of Chinese imperial culture constantly bumped up against, and adapted to, alternative views of the world order which it tended to cover with the rhetoric of universalism: this was its defensive strategy" (Duara 1996, p. 57). According to him, China's political identity exhibits variable and plural boundaries. However, the attribute of "Tianxia" being all-encompassing is demonstrated in the fact that these boundaries can be embraced by "Tianxia". The "external" is, in fact, a conditional "internal". While the boundaries between internal and external may persist for a long time, they are still temporary and incomplete, whereas the universality and permanence of the concept of "Tianxia" are unshakeable. Throughout history, when barbarian tribes in remote regions attained political and military supremacy over the central empire, it heightened the prominence of ethnic boundaries. Nevertheless, this did not shake the cultural preeminence of Confucian cosmopolitanism. Even as the Ming Dynasty approached its downfall and scholars raised the issue of "the collapse of Tianxia", their concerns remained focused on the military setbacks of the empire of Han ethnicity which posed a threat to the survival of Confucian civilization, rather than the complete annihilation or replacement of Confucian universalism by a heterogeneous other. It is precisely this point that faced unprecedented challenges in modern times. This particular impact is a significant reason why Confucian cosmopolitanism is regarded as a political rather than an ethical issue.

## 3. The Modern Predicament of Confucian Cosmopolitanism

Confucianism, as a form of universalism, has faced significant challenges from the impact of Western modernity, necessitating a response. This response has taken various forms: retreating into localized knowledge and being assimilated as the backward "Other" within the framework of Western progress, expanding its denotation to demonstrate Chinese tradition as the source of Western civilization while maintaining its universality, or seeking a comprehensive pluralism to navigate conflicts between competing worldviews. However, this perspective of impact and response has been criticized in both Chinese and Western academia for its inherent Western-centric bias. Zhao Tingyang emphasizes the need for modern China to "rethink China" as a subject of civilization rather than as the other and merely "critiquing China" (Zhao 2011, pp. 1–11). Nonetheless, this process of rethinking must be situated within the historical context.

In modern times, particularly since the late 19th century, the predicament of Chinese culture lies in its transformation from the center of the all-encompassing Tianxia system to the periphery of the world. Confucian cosmopolitanism, as a manifestation of this Tianxia system, has become a marginal and insignificant viewpoint. The rise of prevailing universalistic civilizations based on the nation-state system, Enlightenment values, modern political philosophy, and evolutionism has undermined the premise of "all-encompassing", leaving Confucian cosmopolitanism grappling with a dual predicament.

The first aspect of this predicament is the contradiction between the reality demands of establishing a modern nation-state and the Confucian cosmopolitan ideal that transcends national boundaries. Wang Hui 汪晖 characterizes this conflict as a conflict between the logic of Datong and the logic of national empowerment. The national crisis faced by modern China required prioritizing the nation's salvation, where the political survival of the state became the foundation for confrontations or dialogues between civilizations. Resisting colonial invasion under the banner of nationalism became the common choice of reformers and revolutionaries. However, this path largely affirmed Western modern values, such as constructing a sovereign nation through resistance and following the route of European colonialism and industrialization (Wang 2015, p. 717).

From the late 19th to the early 20th century, nationalism and militarism gained prominence in the intellectual circles of China. Yang Du 杨度 (1875–1931), in his work *Jintie Zhuyi Shuo* 金铁主义说 (*Doctrine of Gold and Iron*), explicitly incorporated this stance in its title. The opening statement of his work reads, "The countries China encounters today are civilized nations, and the world China resides in today is a barbaric world" (Yang 1986, p. 219). Within this context, civilization represents freedom and equality domestically, while externally it requires strength to dominate the weak, reminiscent of barbarism. In order for China to coexist with civilized nations in a barbaric world, Yang advocates for the adoption of economic militarism, which would ensure China's survival and superiority in competition. Yang's discourse establishes a dichotomy between civilization and barbarism, indicating his understanding and plan for the construction of a new China within the framework of modern evolutionism. He did not seek to reclaim or argue for the restoration of Confucian universalism or the concept of Tianxia. Nor did he advocate for the uniqueness of language, customs, or history as the shaping forces of the new nation-state. Instead, he aimed to reshape China through the principles of a free citizenry, responsible government, the prosperity of the people symbolized by "gold", and national strength symbolized by "iron". This approach embodies a practical form of "worldly nationalism".

During this period, Liang Qichao, much like Yang Du, passionately advocated for nationalist ideologies, viewing imperialism, nationalism, and national imperialism as successive stages of social evolution. Liang argued that Westerners utilized social Darwinism to justify the concept of "survival of the fittest" among nations, which he regarded as the driving force behind the rise of modern imperialism (Liang 2018a, p. 695). He believed that China should follow this evolutionary path, engaging in competition with other nations under the banner of nationalism. Although he never advocated for China to colonize other countries, and his discussions on national imperialism focused on devising strategies for national salvation, he did explicitly state that "(China) must first experience the era of nationalism before entering the era of national imperialism" (Liang 2018b, p. 11). It is evident that the rationale behind national empowerment acknowledges the significance of modernity and signifies the failure of Confucian cosmopolitanism. Thus, seemingly, Yang and Liang fell into the trap of colonialism.

However, when one chooses to criticize nationalism and embrace cosmopolitanism, they encounter yet another trap. Following World War I, during the surge of cosmopolitanism promoted by the New Culture Movement, Sun Zhongshan 孙中山 (1866–1925) cautiously maintained that cosmopolitanism could not replace nationalism. He believed that the cosmopolitanism advocated by the new generation of intellectuals, influenced by Britain and Russia and popular in China, was a disguised form of imperialism and aggression. The criticism of nationalism's narrowness and the aspiration for global unity concealed

the conspiracies of imperialist nations seeking to preserve their monopolistic status and impede the resurgence of weaker nations. Sun Zhongshan often used a metaphor to depict China as a poor laborer, nationalism as the carrying pole, his means of livelihood. In contrast, cosmopolitanism was likened to a hidden lottery ticket within the pole. When the laborer discovers he has won the lottery, he rejoices, thinking he no longer needs to sell his labor and discards the pole into the sea. Sun Zhongshan did not dismiss the value of cosmopolitanism, but he believed that, given China's vulnerable situation at the time, cosmopolitanism was merely a lure to deceive the Chinese people into surrendering their resistance. Without grasping the carrying pole of nationalism, China would be unable to preserve its nationhood, let alone achieve the ideal of Datong (Sun 2011, pp. 35–37). Consequently, within the context where Confucian universalism had proven ineffective, Confucian cosmopolitanism found itself in a dilemma: contracting into nationalism would affirm the logic of imperialism and abandon the vitality of the culture, while embracing modern cosmopolitanism seemed to fall into the cunning plot of colonialism, forsaking the vitality of the nation.

The other aspect of this dual predicament lies in the fact that the concepts of Tianxia and Datong have become defensive tools for conservative nationalism. While they may represent an imagined vision of a superior global order, they are primarily perceived as "Eastern" and "ours" before being considered "better". Compared to the previous predicament, the second one may appear less urgent. However, after World War I, when the Western world was no longer unquestionably synonymous with a perfect future, and when Enlightenment rationality and modern values underwent thorough examination and criticism, the voice of cultural nationalism indeed deserved attention and concern. Dongfang wenhua pai 东方文化派" (Eastern Culture Faction) of the May Fourth era embodies these sentiments, emphasizing the differences between Eastern and Western cultures through a comparative lens and firmly believing that the nation can only be saved through the revival of its own culture. According to their perspective, the issues arising from the negative aspects of Western culture can only be resolved by embracing Chinese or Eastern culture. In terms of global order, they argued that Europe's descent into conflict stemmed from Europeans' exclusive focus on the nation-state concept while disregarding the concept of Tianxia.

Liang Qichao was regarded as a prominent figure of this group. In his work *Xinmin Shuo* 新民说 (*New Citizen Discourse*) in 1902, he pointed out that the Chinese people lacked a national consciousness and were "aware of Tianxia but ignorant of nation" (Liang 2018a, p. 546). At that time, he perceived this as a deficiency within Confucian culture. However, after traveling around Europe in the late 1910s, he reassessed his admiration for nationalism and social Darwinism, shifting his focus to China's and particularly Confucian civilization's responsibilities towards world civilization. In his book *Ouyou xinying lu* 欧游心影录 (*Impressions of a Trip to Europe*) he stated, "China was highly developed in terms of the ideal of a united humanity. We have never regarded the nation as the highest human group" and "Formerly, the European conception of Tianxia was not as clear as the Chinese" (Liang 2018c, pp. 155–56). This statement implies that cultural foundations contribute to Europe's division and turmoil. Furthermore, he pointed out that Westerners were fascinated by Confucian teachings such as "all within the four seas are brothers", Mohist principles of "Jianai 兼爱" (universal love) and "Qinbing 寝兵" (cessation of warfare), which led them to seek the incorporation of elements from Eastern civilization. He encouraged young people to respect, reconstruct, and propagate traditional culture because "across the vast ocean, there are several billion people, distressed by the bankruptcy of material civilization, crying out desperately for help, waiting for your deliverance" (Liang 2018c, p. 85).

It was not only the Chinese Cultural Faction that held such a view. In the 1920s, the belief that Chinese culture would save the world was quite widespread. Sun Zhongshan also asserted that the spirit of Chinese pacifism embodied true cosmopolitanism. He recounted an incident during the height of World War I when a British consul tried to persuade him to join the Allies in the war. Sun Zhongshan responded by stating that Chinese civiliza-

tion had progressed over 2000 years beyond Europe, having long abandoned imperialism and advocated for peace. He believed that European warfare was driven by power rather than justice and hoped that China would always uphold the ethics of peace, thus choosing not to participate in the war. Eventually, the British consul agreed with his viewpoint (Sun 2011, pp. 45–46).

The problem with regarding Confucian cosmopolitanism as a superior salvation strategy compared to the West lies not in the pride that Chinese people take in their own culture. The issue arises when the concepts of Tianxia and Datong, characterized by inclusiveness and universality, become part of a binary narrative between Eastern and Western cultures. When these concepts are used to amplify cultural divisions and assert cultural superiority, they undermine their original intentions. Confucian cosmopolitanism should transcend national boundaries and dissolve the distinction between self and the other. However, the excessive emphasis on the historicity and ownership of the doctrine of cosmopolitanism has diluted its universality. It is worth noting that the inclination to assert the superiority of Confucian culture through the cosmopolitan idea of Tianxia and to differentiate groups based on this assertion is not uncommon in Chinese history. However, in the 1920s, China could no longer revert to its pre-modern, self-centered worldview, despite the apparent decline of the Western-centric perspective that had been introduced. Instead, China faced the task of reinventing Confucian cosmopolitanism, striving to enhance the inclusiveness of this worldview that recognizes no external part and is founded on the principle of publicity.

Unsurprisingly, under the dual predicaments, modern nationalism and the nation-state system pose significant challenges and act as direct interlocutors to Confucian cosmopolitanism. Therefore, from its inception, this topic has been a question of political philosophy and practice. Given the pressing and complex external environment, Confucian cosmopolitanism cannot be solely addressed as an issue of the education of individuals, but rather as a problem of institutional construction.[3] However, in navigating these predicaments, modern scholars, particularly Liang Qichao, have not only explored potential modern forms of Tianxia and Datong at the institutional level but have also delved into the inner aspects of human nature and reinterpreted the value of Ren, the fundamental virtue of Confucianism.

## 4. The Reconciliation of the Nation and the World

Confucianism provides its own solution to the first predicament. The theory of three eras proposed by the Gongyang school, mentioned earlier as a way to accommodate the distinction between Yi and Xia, can be slightly modified to assimilate nationalism, perceiving the nation-state as a preliminary phase for the realization of the Datong ideal.

One of the most prominent scholars who applied this theory in modern times is undoubtedly Kang Youwei. In his work *Datong Shu* 大同书 (*Book of Datong*), he vividly depicted the relentless wars and subsequent hardships that arose "once national boundaries were established and national consciousness was born" (Kang 2010, p. 203). His portrayal spanned from the tribal era to the conquest by the Qing Dynasty, even including the historical tribulations of Europe and other Asian countries. He lamented, "How sorrowful! How miserable! All these sufferings are caused by the establishment of national borders!" (Kang 2010, p. 218). While he acknowledged that the annexation of small countries by larger ones is a natural law, he believed that stopping wars and ensuring people's well-being is a matter of "Gongli 公理" (principle) and that Datong is an achievable reality rather than a mere fantasy. In the chapter discussing the "harm caused by the existence of nations", he redefined the theory of three eras, proposing that the realization of the Datong ideal requires three stages, which he still named as "juluanshi 据乱世" (the era of chaos), "shengpingshi 升平世" (the era of stability), and "taipingshi 太平世" (the era of great peace). However, he gave these stages a new definition, starting with an equal alliance among nations, transitioning to a federated world government, and culminating in a "dadi gongyi zhengfu 大地公议政府" (global public government) devoid of individual nations or governors. He

meticulously constructed a "table of the three eras for Datong and the unity of nations", serving as a comprehensive blueprint to achieving Confucian cosmopolitanism. It encompasses various aspects such as representative institutions, judiciary systems, and disarmament processes, as well as transnational transportation and communication. This blueprint goes beyond the traditional conception of the Datong order found in ancient texts. It incorporates modernity by assimilating some of its elements into earlier stages and integrating others into the highest ideal of Datong. Kang Youwei declared the following:

> Within the coming century, all weak nations will undoubtedly face extinction, all autocratic monarchies will be systematically dismantled, and both republican and constitutional governance will be universally adopted. The equality of people's parties will shine brilliantly, and the citizens of civilized nations will universally achieve wisdom, while inferior ethnic groups will gradually fade away. Henceforth, the irresistible momentum of human spirit and societal development will drive us towards global Datong and universal peace, an unstoppable force as potent as water rushing into a chasm. (Kang 2010, p. 226)

Kang Youwei embraced the theory of evolution and engaged with the dichotomous discourse of civilization versus barbarism, even using it to differentiate between races. He viewed the annexation of smaller nations by larger ones as a stepping stone towards the realization of Datong. Kang predicted that within the span of a century, weaker nations would gradually cease to exist, consolidating into a few super-nations on a continental scale. This development would, in turn, establish the groundwork for an international government.

Kang's ideas seemingly exhibit the characteristics of imperialist thinking, sharing a striking resemblance to the logic of national empowerment advocated by Yang Du and Liang Qichao. Furthermore, his portrayal of the third stage—"abolishing nations and territories, establishing autonomous provinces and counties, all united under a public government, akin to the systems of the United States or Switzerland" (Kang 2010, p. 226)—seems to imply that the ultimate goal of Confucian cosmopolitanism resembles an expanded version of the United States or Switzerland. If Kang Youwei's discourse were to end here, he would find himself caught in the first predicament. However, his approach was far more nuanced. Kang aimed to expand and reshape Confucian universalism by incorporating Western theories and experiences into the path towards the Datong ideal. Regarding the Western paradigm of modernization, especially in terms of international relations, Kang placed it within the first stage, the era of chaos. He drew parallels between this era and the dynamics of alliances among powerful states in the Spring and Autumn period, such as Jin 晋 and Chu 楚, as well as the Vienna System in Europe. Simultaneously, Kang Youwei demonstrated an open-minded attitude towards Western modern values, integrating them into his own philosophical framework. He emphasized democracy as a crucial prerequisite for dismantling national boundaries, stating, "When power resides in a monarch, individual self-interest makes unity challenging. Yet, if power rests with the people, unity becomes significantly more attainable. Given that people inherently seek their own advantage, the beneficial concept of Datong, proposed by benevolent individuals, naturally resonates with them" (Kang 2010, p. 221). In addition, he believed that a democratic framework would facilitate a milder merger of countries. According to Kang's vision, a global-scale democratic entity would emerge. However, after the process of unifying nations, ethnicities, and religions, the concept of "nation" or "state" would essentially be discarded. The public government, devoid of any central authority, would consist only of legislators and administrators. Borders, armies, and weapons would be dissolved, while language, transportation, currency, and measurements would be unified. Land, oceans, and taxes would be collectively owned. Religions or deities would not be worshiped nor would there be reverence for the heavens. Instead, respect would be given to the wisdom of the sages and the inherent divinity of each individual. As Wang Hui aptly notes, "It is the persistent interplay, tension, and divergence between the transcendent logic of Datong and the wealth-seeking logic of national empowerment that forms the intrinsic essence of

Kang Youwei's thought" (Wang 2015, p. 747). Kang expanded the scope of the theory of three eras by incorporating individualism, liberalism, and rationalism into the Datong framework, bridging the gap between nationalism and cosmopolitanism. He not only justified China's current need for strength and prosperity within the context of Datong but also enriched and reconstructed the ideal of Datong into a comprehensive knowledge structure that transcends historical boundaries and envisions a global scale.

Kang Youwei's efforts to reshape Confucian cosmopolitanism began quite early, possibly starting works such as *Shili Gongfa Quanshu* 实理公法全书 (the *Complete Book of Public Principles*) and *Datong Shu* as far back as the 1880s. His student, Liang Qichao, had early access to these books. According to Liang, Kang had already worked on these texts when Liang began studying under him in 1891 at the age of 19. Liang learned about the concept of Datong through Kang's discussions with Chen Qianqiu 陈千秋 (1869–1895), an elder student of Kang. In his later works, such as *Qingdai Xushu Gailun* 清代学术概论 (*An Introduction to Scholarship in the Qing Dynasty*) written in 1920, Liang revisited *Datong Shu*. Unlike his initial impression "appreciating its beauty without fully grasping its essence" (Liang 2018b, p. 109), he now had a profound understanding of the modern implications of Kang's adaptation of the theory of three eras. When summarizing the contents of Datong Shu, Liang identified "a world devoid of countries, under a singular global government divided into several regions" (Liang 2018c, p. 275) as the primary point. He regarded this book, written three decades prior, as resembling present-day cosmopolitanism and socialism but more ingenious.

However, Kang and Liang had differing attitudes towards this ideal of Datong, and this discrepancy provides crucial insights into understanding Liang's cosmopolitan propositions. The divergence mainly lies in Liang's belief that, conceptually, the "world" should coexist with the "nation" rather than sequentially replacing it as an evolutionary process. This disparity becomes evident when considering their contrasting opinions regarding the publication of *Datong Shu*. Kang Youwei believed in a progressive sequence of the three eras and argued for keeping the ideal of Datong concealed during the era of chaos, as openly discussing it would create more chaos (Kang 2010, p. 276). He persistently emphasized that the present issues should be addressed with the principles of Xiaokang. In contrast, Liang was eager to widely disseminate the theory of Datong from the moment he encountered it. He repeatedly suggested Kang to publish *Datong Shu*, but his proposals were consistently declined until parts of the book were eventually published in the journal *Buren* 不忍 in 1913. Liang Qichao asserted the following: "The paradox of constructing a new ideal, recognizing it as the epitome of goodness and beauty, yet not desiring its realization and exerting all efforts to resist and hinder it, is the most peculiar phenomenon conceivable" (Liang 2018d, p. 561). After World War I, although Liang conceded that it might be too early to embrace a unified world, he did not support the prevailing global pattern of nations engaging in warfare with each other. He believed that the ideal of Datong, the epitome of goodness and beauty, should not be subtly hidden behind text, like the rhetoric of *Chunqiu*. Instead, Liang raised the banner of cosmopolitanism and engaged in discussions about the current global order with this ultimate aspiration in sight.

In Liang Qichao's work, *Xinqin Zhengzhi Sixiang Shi* 先秦政治思想史 (*A History of Pre-Qin Political Thought*), he conducted a critical evaluation of the cosmopolitan ideas advocated by various Pre-Qin philosophical schools. His central thesis can be stated as follows:

> All Chinese ancient philosophers, regardless of their respective schools of thought, universally considered Tianxia as the subject of their political discourse. Tianxia represents the concept of humanity as a whole. While their understanding of the term "whole" may not have aligned precisely with the modern understanding, these ancient scholars aimed to encompass the broadest human context within their reach, rather than focusing on a limited group. This approach exemplifies the genuine spirit of cosmopolitanism. (Liang 2018d, p. 561)

Although the concept of "Tianxia" discussed by philosophers during China's Axial Age did not objectively encompass all of humanity due to the limitations of their cognitive

scope, it did not hinder their subjective perception of the world as a unified and inclusive entity, and their interpretations of justice and order as universally applicable principles. "The ultimate aspiration of our ancestors was to expand their cultural insights for acceptance and sharing among all of humanity, and establish a platform of equality" (Liang 2018d, p. 561). Furthermore, Liang Qichao argued that during the Pre-Qin era, which was characterized by the coexistence of multiple nations, cosmopolitanism was not a prevailing reality. On the contrary, it was an objective pursued by various schools of thought, who advocated academic theories and propelled cosmopolitanism as a prominent intellectual current of the time. As inter-nation conflicts intensified, especially during the time of Mencius, criticism of nationalism grew among proponents of cosmopolitanism. In this context, Liang's interpretation of *Gongyang Zhuan* on the opening sentence of *Chunqiu*, "Yuannian chun wang zhengyue 元年春王正月" (the beginning year, the spring, the king, the beginning month), offers a distinctive and intriguing perspective:

> In Confucius' *Spring and Autumn Annals*, the opening sentence reads "Yuannian chun wang zhengyue". The *Gongyang Commentary* states, "What does it mean by 'wang zhengyue'? It refers to "dayitong" 大一统 (the grand unification)". The use of the Lu 鲁 state's calendar reflects the prevailing concept of state, which was inherent in the societal norms of that period. However, the significant placement of the character "wang 王" before "zhengyue 正月" carries a profound implication of transcending the boundaries of states. (Liang 2018d, p. 561)

Liang Qichao interpreted "wang" and "zhengyue", respectively, as symbols representing cosmopolitanism and nationalism. The term "zhengyue", tied to the calendar system of the Lu state, reflects the practical expression of national consciousness, while "wang" alludes to the Zhou 周 Emperor, signifying the ruler of the entire world. The prioritization of "wang" over "zhengyue" highlights Confucianism's emphasis on a cosmopolitan order that transcends national boundaries. This interpretation aligns with Liang Qichao's concept of a "cosmopolitan state" as described in *Impressions of a Trip to Europe*.

It is important to note that Liang Qichao's interpretation diverges from the traditional understanding of the Gongyang school and differs from that of his teacher, Kang Youwei. Neither He Xiu 何休 (129–182) nor Kang Youwei considered "zhengyue" to represent the Lu state's calendar. He Xiu interpreted "zhengyue" as a calendar reform ordained by the king and mandated by heaven. Kang Youwei proposed that the original text of *Chunqiu* before Confucius made revisions, namely "buxiu Chunqiu 不修春秋", recorded "Yinian chun yiyue 一年春一月" (the first month of spring in the first year). Confucius added the character "王" and modified "一" to "正", highlighting the authority and mandate of a king in establishing the calendar (Kang 2016, pp. 45–52). Thus, these interpretations did not explicitly acknowledge the concept of a nation or state. Even the metaphorical depiction of "wang lu 王鲁" (regarding Lu as the king) is more of a rhetorical construct, representing an ideal universal order reformer rather than recognizing Lu as a legitimate "state".

Furthermore, both He Xiu and Kang Youwei mentioned the idea of "using the governance of the king to rectify the accession of feudal lords". The position of feudal lords, whether or not they can be considered "states", is derived from the governance of the king who received the mandate from heaven. The term "wang" in this context is explained in *the Gongyang Zhuan* as "wenwang 文王". He Xiu interpreted this as a reference to King Wen of Zhou 周文王, the earliest heaven-mandated sovereign of the Zhou Dynasty. Kang Youwei, however, took a more radical approach, seeing "wenwang" as "wenming zhi wang 文明之王" (the king of civilization), which he identified with Confucius himself. This interpretation enhanced Confucius's status as a preeminent legislator. In both cases, the interpretation of "wang Tianxia 王天下" (be the king of the world) or "dayitong" consistently emphasizes the absolute, unique, and supreme nature of the cosmic order. This order not only holds greater legitimacy compared to nationalism but also carries ontological validity rooted in heaven and transmitted by sages and kings. It is within this prescribed order that states and feudal lords exist.

Liang Qichao's interpretation, however, aims to strike a balance between global and national perspectives by expanding the national framework to incorporate a broader global outlook. This idea gained significance in the aftermath of World War I, where the nation-state remained dominant in international politics, but the League of Nations emerged as a point of contention. Liang advocated for the League, not solely to protect China's interests, but as a starting point for reconciling global and national aspirations. He emphasized its role in curbing excessive national ambition and promoting moderation. His depiction of a "cosmopolitan state" encapsulates this idea: "Our patriotism must recognize not only the nation but also the individual, and similarly, acknowledge the world in addition to the nation. Our goal is to seek protection under this nation, to maximize individual potential within the country, and to make significant contributions to the global civilization" (Liang 2018c, p. 71). This passage highlights two key themes. Firstly, it underscores the limitations of the nation and recognizes the existence of individuals and a broader global community. The nation, in Liang Qichao's perspective, serves as a means to an end, with the ultimate value lying in the world rather than merely within national interests. Secondly, it emphasizes the significance of individuals. In his broader works, Liang Qichao distinctly emphasized the value and ethics of individuals, providing a pathway to confront the second predicament: the limitation imposed by cultural nationalism that appears to be inherent in Confucianism. Moreover, his perspective serves as a bridge that connects the institutional world with the ethical world. Remarkably, Liang's modern interpretation of "Datong" aligns closely with contemporary English scholarly discussions on Confucian cosmopolitanism.

### 5. The Decentralization of Datong

As discussed earlier, Liang Qichao once proclaimed China's cosmopolitanism as unique and superior. Such a belief can easily lead to cultural arrogance and confine cosmopolitanism within a narrative centered around the nation, ultimately falling back into the trap of nationalism. To overcome this limitation, Liang Qichao sought to fully develop the ideals of Datong and Tianxia through the concept of "tonglei yishi 同类意识" (kinship consciousness). He shifted the focus from the nation to individuals and personal character, decentralizing and internalizing the vision of Datong. The mission of achieving Datong is no longer limited to a specific country or civilization, but rather becomes an inherent aspect of personal growth and perfection.

In his work *A History of Pre-Qin Political Thought*, Liang Qichao elaborated on the concept of Datong:

"Datong" represents the ultimate realization of a complete and harmonious human personality within the universe. However, the universe is never completely perfect; if it were, it would no longer be a universe. Confucians deeply hold this principle, as reflected in the sixty-four hexagrams of *Yi* 易 (the *Book of Changes*), starting with "Qian 乾" and ending with "Weiji 未济". In this imperfect universe, our task is to continuously progress based on our capabilities, inching closer to the realization of the ideal personality and the harmonious universe we aspire to. How can we achieve this? By expanding our kinship consciousness to its utmost extent. However, many people remain numb and unaware of this consciousness. Thus, before discussing further expansion, the first step is to awaken this consciousness. The first step towards awakening begins with the simplest and closest ''xiang ren ou 相人偶" (reciprocal relationships). For instance, recognizing that a father's role is fulfilled by his son, and a son's role by his father; or a husband's role by his wife, and a wife's role by her husband. By understanding these reciprocal human relationships, we can then contemplate expanding this consciousness to a broader scale (Liang 2018d, p. 483).

While the concept of a "cosmopolitan state" considers cosmopolitanism as a complement to nationalism, advocating that the notions of the world and the state, the logic of Datong and national empowerment, should coexist, Liang Qichao consistently perceived the Confucian ideal of global order, Datong, as the ultimate aim of history. This lofty and

unattainable goal illuminates the path towards it, which is underscored by the awakening and refinement of "kinship consciousness". This concept originates from Liang Qichao's interpretation of Confucian personalism. During this period, Liang Qichao enthusiastically emphasized the significance of personalism, asserting that "In Confucianism, without the philosophy of life, there is no true learning, and without personalism, there is no philosophy of life" (Liang 2018d, p. 479). The concept of kinship consciousness is closely intertwined with the core Confucian value of Ren. In essence, they are virtually synonymous. Kinship consciousness is centered on one's recognition of the categorical concept of 'humanity' and the shared similarities between self and the other. Zheng Xuan 郑玄 (127–200), a renowned classical scholar from the Han Dynasty, interpreted "Ren" as "xiang ren ou", which implies reciprocal relations with others. Building on this interpretation, Liang Qichao proposed that kinship consciousness is the conscious recognition of one's shared existence and mutual interdependence with others, which is essential for the formation of personhood. In this context, both kinship consciousness and Ren lead to a unity of self and the other. Liang Qichao further claimed that Ren is synonymous with the awakening of kinship consciousness, presenting a fresh reinterpretation of Ren through the lens of kinship consciousness, emphasizing the concept of mankind.

He contended that the formation of the concept of mankind manifests as kinship consciousness on the cognitive dimension and as empathy on the emotional dimension—a representation of love and care for one's own kind. The negative expression of this emotion is captured in the principle of "Shu 恕", which urges individuals not to do unto others what they do not wish for themselves (己所不欲勿施于人). Conversely, its positive manifestation is described as "aspire to establish people when you wish to establish yourself, aspire to succeed people when you wish to succeed yourself" (己欲立而立人, 己欲达而达人), which characterizes Ren. Intriguingly, Liang Qichao emphasized that the word "ren 人" one aspires to assist is not another individual but humanity as a whole:

> The position I desire to attain presently must be reached in cooperation with my fellow humans. The position I aspire to reach in the future must be advanced together with my fellow humans. Why is that? Human life thrives on reciprocal relationships (that is, "xiang ren ou"). Without everyone collectively establishing this position, I cannot possibly achieve it independently; without everyone jointly reaching this status, I cannot attain it solely. "Establish people" and "succeed people" do not merely imply aiding other individuals but refer to facilitating the success of mankind. As the 'other' and I collectively constitute humanity, facilitating their success equates to advancing humanity, and in doing so, I advance myself as well. (Liang 2018d, p. 478)

Liang Qichao notably emphasized the positive and creative aspects of empathy. However, the principle expressed by "aspire to establish people when you wish to establish yourself" faces an inherent issue: it can potentially lead to the hegemony of universalism, where one imposes their will on others, denying the subjectivity of others. This criticism is applicable to both modern imperialism, which often disguises its aggression under the pretext of a civilizing mission, and the notion that Chinese culture would be the savior of the world. However, Liang's reinterpretation specifically addresses this issue. Instead of interpreting the phrase as enabling others to succeed before oneself or extending one's desired benefits to others, he emphasized the unity of self and the other. This unity implies that personal achievements must be based on recognizing and including all of humanity, and that the shared accomplishment of humanity as a whole is essential.

Undoubtedly, Neo-Confucianism's ontology, which posits that "Ren signifies embracing the cosmos and all living beings as one entity" (仁者, 以天地万物为一体), has long acknowledged the holistic and inclusive nature of Ren. Liang Qichao's explanation incorporates the metaphor of "shou zu bu ren 手足不仁" (insensitivity of limbs) from Cheng-Zhu Neo-Confucianism, where a lack of empathy for others is likened to the numbness of not perceiving one's own limbs. However, Liang's innovation lies in connecting Confucian



moral philosophy with political philosophy, intertwining "Ren" with "Tianxia" and binding kinship consciousness with cosmopolitanism.

Liang Qichao argued that Confucian political theory is deeply rooted in the philosophy of life. After reinterpreting the moral value of Ren, he shifted his focus to its political implications, stating that "Zheng zhe, zheng ye 政者, 正也" (to govern is to rectify). According to Liang, Confucian political aspirations revolve around Tianxia, but governance itself is not the primary concern. Instead, the emphasis lies on achieving harmony. Confucian ideals do not aim to establish a uniform world governed by a specific political system but rather to create a world characterized by "ping" (平, harmony) and "zheng" (正, rectitude). These concepts are deeply rooted in the "jieju zhi dao 絜矩之道" (the way of measuring and aligning), which refers to the kinship consciousness that involves perceiving others through empathy, understanding emotions through sympathy, and recognizing similarities through categorization. The origin of this kinship consciousness can be found in the love we have for those closest to us, which gradually extends from the family to the nation and eventually to other nations. Therefore, achieving harmony and rectitude in Tianxia means expanding this kinship consciousness to its fullest extent. The essence of Tianxia and Datong lies in the comprehensive development of this kinship consciousness and the ultimate refinement of every individual's personality. Liang avoided attributing this process to any specific political entity and did not portray the nation as the central actor. Since the ultimate goal is for every individual in the world to adopt a global perspective, take responsibility for the world, and become an integral part of humanity as a whole, the subjectivity of each individual should indeed be emphasized.

It is undeniable that Liang Qichao believed in the role of Confucianism as an educator in awakening and expanding kinship consciousness. He argued that the reason ancient China was a unified country rather than fragmented into numerous states like Europe was largely due to the promotion of kinship consciousness by Confucianism. Liang asserted, "While external factors certainly played a significant role in our unity, the foremost reason lies in the transformative power of the sage's teachings, which unifies the minds of the majority. I staunchly affirm that kinship consciousness should be broadened, not limited. If our ancestors had consistently fostered a love only for one's own state, such as the people of Qin 秦 loving Qin and the people of Yue 越 loving Yue, the divergent national characteristics between Qin and Yue would be as substantial as the differences between Germany and France today" (Liang 2018d, p. 563). Moreover, he indeed regarded the recognition of kinship consciousness as a crucial distinction between "us" and "them". He stated, "We emphasize the cultivation of our kinship consciousness, leading to convergence despite differences, whereas they prioritize exploiting their differences, resulting in further divergence" (Liang 2018d, p. 563). In this context, "we" represents the Confucians, while "they" refers to Western philosophers. Based on his concept of kinship consciousness, Liang Qichao argued that Confucian political thought differs significantly from prevalent Western ideologies. He strongly criticized nationalism, as it tends to idealize narrow forms of patriotism and view other nations as adversaries. This sentiment aligns with Mencius' idea of "beginning with what they do not care for, and proceeding to what they care for" (以其所不爱及其所爱), which ultimately entangles both their own nation and others in the perils of warfare. Furthermore, Liang extended this critique to socialism, acknowledging that although socialists show empathy towards the working class, they advocate for a contradictory approach of imposing on others what they themselves do not desire. By perpetuating class divisions and considering capitalists as outsiders, they inadvertently hinder the expansion of kinship consciousness.

However, taking a different perspective, Liang Qichao, with the concept of kinship consciousness that illustrates the disparities between Chinese and Western ideologies, does not necessarily propose replacing the West with distinct Chinese systems, values, or cultural traits. Instead, his viewpoint seeks to dissolve the distinction between "we" and "they", China and the West. Formally, Liang employed the language of Confucianism in his discourses and might reinforce the barriers between the self and the other. Neverthe-

less, the essence of his discourse negates these barriers and emphasizes the shared humanity and similarities between individuals. In other words, the Confucian cosmopolitanism he advocated does not necessarily negate its inherent premise of inclusiveness and all-encompassing simply because it is associated with Confucianism. On the contrary, it is possible to transcend the limitation of being solely "Confucian" by focusing on the awakening and refinement of human kinship consciousness. Around 1920, Liang Qichao frequently used the term "the whole of humanity" (人类全体) to contemplate future issues on a global scale. According to his perspective, various groups of people, including nations, serve as means to organize and facilitate the development of mankind, representing stages in the expansion of kinship consciousness. However, they are temporary and limited, while only the "self" has a direct connection to the whole of humanity. The whole of humanity is the maximum extension of the self, with the self serving as the foundation of the whole of humanity. Therefore, the expansion of the "self" towards "the whole of humanity" in each individual has come to embody the modern significance of the ideal of Datong. This profound meaning allows Confucian cosmopolitanism to transcend ideological constraints and decentralize and eventually surpass the confines of the "Confucian" discourse.

## 6. Conclusions

From an intellectual historical perspective, it becomes evident that within the Chinese-speaking sphere, the discourse on cosmopolitanism in modern times has been deeply entwined with the nation-state system and the rise of nationalism. The worldview of Tianxia characterized by its all-encompassing nature and the ideal of Datong based on the principle of publicity have become obsolete, which forms the backdrop for the modern predicament of Confucian cosmopolitanism. In a world marked by boundaries and fragmentation, the modern challenge of Confucian cosmopolitanism lies in reconciling the conflicting logics of Datong and national empowerment, as well as balancing Confucian identity with global concerns.

Through the study of Liang Qichao and his contemporaneous scholars, we find a potential pathway to address these dual predicaments. On one hand, Liang made significant efforts to understand and embrace the modern context, putting forth the concept of a "cosmopolitan nation" that acknowledges the coexistence of the world and the nation. He sought a balance between the ultimate ideal of Datong and the practical reality of nationalism. While recognizing the limitations of the nation-state system, he advocated for a nation that embodies the principles of cosmopolitanism, regarding it as an organizing and mobilizing method to realize this goal. On the other hand, Liang critically reevaluated the values of Confucianism in light of modernity. He offered a fresh interpretation of the ideals of Datong and Ren, bridging Confucian ethics and political theory through the notion of kinship consciousness. His aim was to restore the essence of Confucian cosmopolitanism as the awakening and refinement of human kinship consciousness. This interpretation not only challenged modern ideologies such as nationalism and socialism but also transcended the confines of the Confucian discourse. Ultimately, the focus on individual values directs attention towards addressing human concerns from a global perspective.

As of today, the Chinese-speaking academic community has once again sparked a fervor in exploring the concept of Tianxia, giving rise to theories like "New Tianxiaism" (see Liu 2015; Xu 2015). The practical concern behind this trend is to reconstruct China's understanding of the world order. This reflects that the Chinese cosmopolitanism is still influenced by modern predicaments from the past century, but it has undergone some changes. China consciously seeks a development path consistent with the logic of Datong, offering an alternative proposal different from Western modernity, which appears to alleviate the first modern predicament. However, amidst the current prevalence of nationalism, populism, and anti-globalization sentiments, the second predicament, namely cultural nationalism, is accentuated.

Liang Qichao posited China's responsibility towards world civilization as a process of "integration": "What we should do is to enrich our civilization with Western civiliza-

tion and, at the same time, contribute our civilization to supplement Western civilization, so that the two can integrate into a new civilization" (Liang 2018c, p. 83). While this was an attempt to transcend the confines of cultural identities, in today's world characterized by a more diverse global landscape, the scope of integration should be expanded. The discussion on cosmopolitanism must move beyond the Western–Chinese dichotomy and embrace a global perspective. For instance, Hindu philosophy also offers the concept of "Vasudhaiva Kutumbakam" (the cosmos is one family), which promotes a humanistic view that seeks to dissolve boundaries between self and others (see Ranganathan 2015; Hatcher 1994). In conclusion, as the Daoism motto goes, "seeing Tianxia through the view of Tianxia" (以天下观天下), the global landscape requires acknowledging and appreciating diverse cultural heritages, fostering extensive intercultural dialogues to seek commonalities and embrace diversities. Such an approach may serve as a potential pathway to overcome the modern predicaments of cosmopolitanism.

**Funding:** This research received no external funding.

**Data Availability Statement:** Not applicable.

**Conflicts of Interest:** The author declares no conflict of interest.

## Notes

[1]   Regarding the discussion on cosmopolitanism within the Chinese context, see Ma and Sun (2014), Liu (2016), and Wang (2020).

[2]   The translation of the chapter Liyun of this paragraph and the subsequent text is based on the work of James Legge (Legge 1885, pp. 364–72). To ensure terminological consistency in this article, minor modifications have been made to certain statements, such as replacing "the Grand Union" with "Datong".

[3]   Undoubtedly, the term "shijie zhuyi 世界主义", as the translation for "cosmopolitanism", in the Chinese world has other specific reasons for its strong political implication. For example, internationalism, which has a stronger political connotation and emphasizes a world order transcending national boundaries, is not equated with "shijie zhuyi" but translated as "guoji zhuyi 国际主义", yet this term has its own particular referent, that is, the international alliance formed by the proletarian states advocated by Marxism (see Chen 2021). My article focuses on the awareness of modernity and the modern transformation process of Confucian world ideals, so the issue of translation and dissemination will not be discussed here.

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
