# Peer review of "Confucian Cosmopolitanism: The Modern Predicament and the Way Forward"

_religions, doi:10.3390/rel14081036_

Round 1

Reviewer 1 Report

This is a worthwhile contribution to the literature of Chinese cosmopolitanism and shows sound scholarly practice.

Is content succinctly described? Needs to add to Western counterpart of cosmopolitanism two important sources: one classical, Diogenes of Sinope, who claimed to be “a citizen of the world”; the other contemporary Western liberal cosmopolitan theory based on Emmanuel Kant.

Arguments and discussion of findings? Line 83, referencing Zhao 2011, states that Tianxia ‘represents an “institutionalized world” that encapsulates the full concept of the world’. As I see it, 'institutionalized world' is the 'li' (, propriety), is the outward order-making representation of the fundamental 'ren' (, benevolence). So Tianxia is more political but cannot quite measure up to 'full concept of the world', as that goes beyond the institutional and includes pluralism and context – as shown, for example, in yi (), appropriateness to a particular circumstance. 

Are conclusions thoroughly supported: Can be improved by strengthening the Conclusion. What about non-Western and non-Chinese philosophy and culture? A statement of global focus rather than a Western-Chinese dichotomy and synthesis would be more relevant to the earlier mentioned current Chinese government slogan of a 'community of shared future for mankind' (line 44). A more encompassing statement about the global reach of Tianxia would also overcome the disjuncture between Liang's historical times and the present. This is also a reason not to end with Liang’s own words but to expand on the quote by the author’s elaboration, so that Tianxia does have conceptual integration between China and the West (as argued), but also extends to conceptual understandings of others. Hence the importance of better addressing pluralism and context, as I noted above. Moreover, such a concluding strategy would better address the promise of a ‘way forward’ in the title.

Author Response

  1. I have included an overview of cosmopolitanism in Western philosophy at the beginning of the article, introducing the multifaceted nature of cosmopolitanism:In the Western philosophical tradition, "cosmopolitanism" is a multifaceted and richly nuanced topic with roots in various historical sources. From Diogenes' concept of "kosmopolitēs" to the Stoic idea of a "world city-state" and Kant's vision of a "league of nations," each of these intellectual origins sheds light on different issues. In contemporary philosophical discussions, the exploration of cosmopolitanism has also diversified, giving rise to varied perspectives. For many philosophers, cosmopolitanism is regarded as an ethical matter, sparking debates about whether individuals should extend their care and assistance to all human beings without distinction or if they have specific duties primarily to their fellow compatriots. On the other hand, a significant number of thinkers view cosmopolitanism as a pressing political philosophical issue, grappling with questions about the most just and equitable form of global political organization. (see Scheffler 1991, Kleingeld and Brown 2009)
  2. The concept of "Tianxia" has a wide range of meanings, and in Zhao Tingyang's discussion, he divides the fundamental meanings of Tianxia into three layers: (1) the geographical sense of "all the land under heaven"; (2) the collective mindset of all people living on the land, known as "minxin"(民心) or the will of the people, and in the Chinese political tradition, "de Tianxia" (得天下)means obtaining the support and approval of the people; (3) most importantly, the ethical/political sense of "Tianxia." The phrase "an 'institutionalized world' that encapsulates the full concept of the world," which I cited in my article, comes from this context. His theory of the "Tianxia system" is based on this understanding of Tianxia. Although Zhao Tingyang's discussion of Tianxia leans toward political issues, he emphasizes the holistic nature of the concept, stating that Tianxia represents a philosophy and worldview that forms the foundation for understanding the world, things, people, and cultures. Tianxia constitutes the true foundation of Chinese philosophy, directly defining a philosophical perspective where the object of thought—the world—must be expressed as a complete or meaningful concept that is systematic and comprehensible.

    In my view, although the emphasis on the system may make Tianxia appear similar to the concept of "Li" and be seen as a political issue, in fact, the concept of Tianxia is much broader and not solely political in nature. Firstly, it is essential to clarify that the Confucian understanding of Li is not purely external; it is rooted in Ren, and the interpretation of the moral content and educational purpose of Li, known as "Li Yi(礼义)," is an essential aspect of the study of Li. Moreover, as an ideal world vision, Tianxia may not necessarily match the reality of Li as a system design. For example, Kang Youwei distinguishes between the concepts of "Li Yun" (礼运) and "Ren Yun" (仁运); Ren Yun represents the way of Datong, while Li Yun represents a more practical approach. Tianxia is an expectation of a systematic and comprehensible order in the world, rather than a fixed set of systems or answers. Furthermore, the concept of Tianxia indeed has its cosmological and metaphysical foundation, stemming from the Confucian understanding of heaven and closely connected to theories of human nature and ethics. Therefore, to support a modern interpretation of Tianxia, a greater emphasis on the concept of Ren is needed.

  3. I highly appreciate your suggestions regarding the conclusion of the article, and I have made the following modifications as per your advice:

    As of today, the Chinese-speaking academic community has once again sparked a fervor in exploring the concept of Tianxia, giving rise to theories like "New Tianxiaism." (see Liu 2015, Xu 2015). The practical concern behind this trend is to reconstruct China's understanding of the world order. This reflects that the Chinese cosmopolitanism is still influenced by modern predicaments from the past century, but it has undergone some changes. China consciously seeks a development path consistent with the logic of Datong, offering an alternative proposal different from Western modernity, which appears to alleviate the first modern predicament. However, amidst the current prevalence of nationalism, populism, and anti-globalization sentiments, the second predicament, namely cultural nationalism, is accentuated. 

    Liang Qichao posited China's responsibility towards world civilization as a process of "integration": "What we should do is to enrich our civilization with Western civilization and, at the same time, contribute our civilization to supplement Western civilization, so that the two can integrate into a new civilization" (Liang 2018c, p.83). While this was an attempt to transcend the confines of cultural identities, in today's world characterized by a more diverse global landscape, the scope of integration should be expanded. The discussion on cosmopolitanism must move beyond the Western-Chinese dichotomy and embrace a global perspective. For instance, Hindu philosophy also offers the concept of "Vasudhaiva Kutumbakam" (the cosmos is one family), which promotes a humanistic view that seeks to dissolve boundaries between self and others. (see Ranganathan 2015, Hatcher 1994). In conclusion, as the Daoism motto goes, "seeing Tianxia through the view of Tianxia"(以天下观天下), the global landscape requires acknowledging and appreciating diverse cultural heritages, fostering extensive intercultural dialogues to seek commonalities and embrace diversities. Such an approach may serve as a potential pathway to overcome the modern predicaments of cosmopolitanism.

Reviewer 2 Report

This is an excellent contribution to contemporary chinese philosophy. The author engages with the concept of Confucian cosmopolitanism and tries to understand how nationalism can coexist with the ideal of cosmopolitanism and how individuality is related to community by discussing Liang Qichao and other chinese Confucian philosophers. This is a well-written paper and an original contribution to the contemporary discussion in chinese political philosophy. The objectives of the paper are clearly articulated and the analysis is methodologically adequate.

Some minor issues:

(1) The author should explain more clearly the term "intersubjectivity" (p. 12, line 588) and how this term is related to "kinship consciousness".

(2) The author should add more bibliography to the paper.  

Author Response

1. The term "intersubjectivity" used here does not strictly adhere to Husserl's or Heidegger's definitions. Instead, it refers to the shared existence and mutual interdependence between individuals, dissolving the binary relationship between self and others. Recognizing this connection is akin to awakening to a sense of solidarity.

Given the multiple interpretations of intersubjectivity across different fields, using this concept in the text may lead to misunderstandings. Explaining the precise meaning of this term in the context would disrupt the flow of the writing. Moreover, introducing the concept of intersubjectivity is unnecessary in this passage. Hence, I have opted to replace it with simpler language, resulting in the revised paragraph:

Building on this interpretation, Liang Qichao proposed that kinship consciousness is the conscious recognition of one’s shared existence and mutual interdependence with others, which is essential for the formation of personhood.

2. I have added the following references to the bibliography:

Hatcher, Brian A. 1994. “The Cosmos is One Family”(Vasudhaiva Kutumbakam): Problematic Mantra of Hindu Humanism. Contributions to Indian sociology 28(1): pp.149-162.

Kleingeld, Pauline and Eric Brown, "Cosmopolitanism", The Stanford Encyclopedia of Philosophy (Winter 2019 Edition), Edward N. Zalta (ed.), URL = https://plato.stanford.edu/archives/win2019/entries/cosmopolitanism/

Liu, Qing 刘擎. 2015. Reimagining the Global: From 'Tianxia' to New Cosmopolitanism 重建全球想象:从“天下”理想走向新世界主义, Academic Monthly 学术月刊 47(8): pp.5-15.

Ranganathan, Ramya. 2015. Vasudhaiva Kutumbakam (The World is my Family): What Happens to My Self-concept When I Take Others' Perspectives? South Asian Journal of Management 22(4): p.118.

Scheffler, Samuel. 1999. Conceptions of Cosmopolitanism. Utilitas, 11(3): pp. 255-276.

Xu, Jilin 许纪霖. 2015. New Tianxiaism and China's Internal and External Order新天下主义与中国的内外秩序. In New Tianxianism新天下主义. Edited by Xu Jinlin许纪霖 and Liuqing刘擎. Shanghai上海: Shanghai Renmin Press上海人民出版社, pp.3-25.